# Management of Upper-Limb Spasticity Using Modern Rehabilitation Techniques versus Botulinum Toxin Injections Following Stroke

**DOI:** 10.3390/life13112218

**Published:** 2023-11-17

**Authors:** Ana Maria Bumbea, Otilia Constantina Rogoveanu, Adina Turcu-Stiolica, Ionica Pirici, George Cioroianu, Diana Iulia Stanca, Oana Criciotoiu, Viorel Biciusca, Rodica Magdalena Traistaru, Danut Visarion Caimac

**Affiliations:** 1Department of Physical Medicine and Rehabilitation, University of Medicine and Pharmacy of Craiova, Petru Rares 2, 200349 Craiova, Romania; anamaria.bumbea@umfcv.ro (A.M.B.); otilia.rogoveanu@umfcv.ro (O.C.R.); 2Department of Pharmacoeconomics, University of Medicine and Pharmacy of Craiova, Petru Rares 2, 200349 Craiova, Romania; adina.turcu@umfcv.ro; 3Department of Anatomy, University of Medicine and Pharmacy of Craiova, Petru Rares 2, 200349 Craiova, Romania; 4Doctoral School, University of Medicine and Pharmacy of Craiova, Petru Rares 2, 200349 Craiova, Romania; cioroianu_geroge@yahoo.com; 5Department of Neurology, University of Medicine and Pharmacy of Craiova, Petru Rares 2, 200349 Craiova, Romania; diana.stanca@umfcv.ro; 6Department of Internal Medicine, University of Medicine and Pharmacy of Craiova, Petru Rares 2, 200349 Craiova, Romania; viorel.biciusca@umfcv.ro; 7Medical Rehabilitation Department, Nursing Faculty, University of Medicine and Pharmacy, Petru Rares 2, 200349 Craiova, Romania; rodica.traistaru@umfcv.ro (R.M.T.); danut.caimac@umfcv.ro (D.V.C.)

**Keywords:** stroke, spasticity, botulinum toxin, upper limb

## Abstract

Our purpose is to emphasize the role of botulinum toxin in spasticity therapy and functional recovery in patients following strokes. Our retrospective study compared two groups, namely ischemic and hemorrhagic stroke patients. The study group (BT group) comprised 80 patients who received focal botulinum toxin as therapy for an upper limb with spastic muscle three times every three months. The control group (ES group) comprised 80 patients who received only medical rehabilitation consisting of electrostimulation and radial shockwave therapy for the upper limb, which was applied three times every three months. Both groups received the same stretching program for spastic muscles as a home training program. We evaluated the evolution of the patients using muscle strength, Ashworth, Tardieu, Frenchay, and Barthel scales. The analysis indicated a statistically significant difference between the two groups for all scales, with better results for the BT group (*p* < 0.0001 for all scales). In our study, the age at disease onset was an important prediction factor for better recovery in both groups but not in all scales. Better recovery was obtained for younger patients (in the BT group, MRC scale: rho = −0.609, *p*-value < 0.0001; Tardieu scale: rho = −0.365, *p*-value = 0.001; in the ES group, MRC scale: rho = −0.445, *p*-value < 0.0001; Barthel scale: rho = −0.239, *p*-value = 0.033). Our results demonstrated the effectiveness of botulinum toxin therapy compared with the rehabilitation method, showing a reduction of the recovery time of the upper limb, as well as an improvement of functionality and a reduction of disability. Although all patients followed a specific kinetic program, important improvements were evident in the botulinum toxin group.

## 1. Introduction

Stroke is one of the main causes of mortality and disability in surviving patients worldwide. More specifically, stroke is the second highest cause of morbidity and mortality, and motor deficit is the third most common sequela found in stroke patients [1,2].

Thus, stroke remains a health problem worldwide. This assertion is supported by statistical data that are worrying regarding mortality and residual disability after a stroke. In the European Union in 2017, there were 1.12 million cases of stroke, resulting in 0.46 million deaths and 7.06 million patients with disabilities who required additional medical care, personal caretakers, and auxiliary medical devices, such as orthoses and wheelchairs, to improve quality of life. By 2047, it is estimated that there will be a 3% increase in case incidence, a 27% increase in prevalence, a 17% decrease in mortality rate, and a 33% decrease in mortality compared to present figures. The decrease in mortality rate is estimated to be lower for less-developed countries, such as Romania, where the estimated mortality rate decrease is only 0.23%. Romania is one of the top three countries in terms of stroke cases, death, and disability [3,4].

Stroke is the second highest cause of death on a world scale, the same as in Romania, with an increasing trend in incidence and prevalence globally, so it is estimated that by 2030, it will be the main cause of death worldwide. Surviving patients, estimated to be an increasing population, will have a permanent disability, according to the extent of the stroke, for the rest of their lives. This aspect of permanent disability, with great effects on the life quality of the patient and their family, makes this disease a major health problem [5].

In Romania, stroke prevalence is 252,774 cases per year, with a rate of 8333 cases per 100,000 inhabitants, which represents a very high rate and explains the interest in finding new therapeutic solutions to minimize the disability through combined pharmacological and rehabilitation techniques [6].

The WHO reports that stroke is the second highest cause of death in Romania, after heart attack, with a very small difference between the sexes [7]. Stroke is defined as rapidly developing clinical signs of focal or global disturbance of cerebral function lasting more than 24 h or leading to death with no other origin than vascular. In more than 60% of strokes, there are symptoms related to spasticity. The clinical characteristics of spasticity are high tone, hyperreflexia, flexor spasm clasp knife reaction, extensor spasm, and associated reactions [8].

In stroke patients, there are several stages of evolution. In the early stage, patients typically exhibit motor deficits, abolished tendon reflexes, and the appearance of pathological specific reflexes. Swallowing deficit, sphincter control deficit, impaired speech, and cognitive disorders may also be observed. The spastic phase begins after a variable time, usually within a few weeks of the onset of stroke. Spasticity affects specific muscle groups, such as the flexors of the upper limbs and the extensors of the lower limbs. The arm tends to assume a pronated and flexed position, and the leg assumes an adducted and extended position. These positions indicate that some spinal neurons are reflexively more active than others. There is no constant relationship between spasticity and weakness. The pathophysiology of spasticity is further dependent on two descending tracts: the dorsal reticulospinal tract and the medial reticulospinal and vestibulospinal tracts. The dorsal reticulospinal tract has inhibitory effects on stretch reflexes. Medial reticulospinal and vestibulospinal tracts facilitate the extensor tone. This is the moment when reflexes intensify, and it is also the ideal moment to begin rehabilitation. For adequate rehabilitation, spasticity must be kept at an appropriate level to initiate and continue rehabilitation [9].

Spasticity is a disorder of the stretch reflex that is clinically manifested by increased muscle tone [10,11]. Also, spasticity is a common condition in post-stroke patients that can be associated with pain and joint contracture [12,13], which leads to decreased quality of life through vicious limb positions, deformity, involuntary movement, and medical complications (skin maceration and pressure sores) when untreated [14]. Spasticity after stroke occurs in approximately one third of patients and has been shown in many studies to have a negative effect on a patient’s life and influences upper-limb function negatively [15], which can lead to falls, fractures, and a difficult recovery [16].

The motor control of the affected limb being deficient causes abnormal movements, misdirected systematically, which is a primary consequence of brain injury and a secondary non-use consequence [17].

Reducing disability and recovery time is an increasingly important aspect nowadays, given the high costs and socioeconomic implications. Thus, finding new therapeutic methods to reduce the remaining spasticity becomes a major objective. Spasticity management is a complex mechanism that requires a holistic approach which includes pharmacological therapy associated with adequate and personalized rehabilitation programs. The objective of spasticity treatment is to reduce the motor hyperactivity and improve mobility, but without accentuating the motor deficit [18].

The advantages of local therapy over other spasticity treatments are that unlike the systemic anti-spasticity drugs which are commonly associated with generalized weakness and functional loss, botulinum toxin is a targeted therapy and unlike chemical neurolysis with alcohol or phenol injection does not causes skin sensory loss or dysesthesia [8].

The pharmacological treatment for spasticity in stroke patients includes both focal, localized administration of medication in the spastic muscle and also conventional oral therapy. Systemic therapy distributes medication throughout the body, without specifically targeting the spastic muscle, making it less beneficial for patients. On the other hand, focal therapy involves injecting botulinum toxin directly into the spastic muscle, the target zone of treatment, using ultrasound-guided in situ injection with a precise and personalized dosage, for each muscle group, every 3 months or more [14,16].

The objectives of our study were to highlight the differences between botulinum toxin type A (incobotulinum toxin and abobotulinum toxin) and recovery therapy, combined with specific kinetic programs, in the management of spasticity and functionality in stroke patients. We aimed to emphasize that the association of a kinetic program with focal therapy with botulinum toxin leads to better results compared to those of the group that received the same kinetic program but with electromyostimulation and radial shockwaves. This combination proves more effective in enhancing muscle force and functionality and reducing spasticity to a convenient level.

## 2. Materials and Methods

The presented retrospective study was conducted over a period of three years, from September 2020 to September 2023, in the Neurorehabilitation Department of the Neuropsychiatry Hospital in Craiova, Romania. The study followed the treatment protocols for stroke patients accepted in our department.

The patients gave their informed consent according to the Declaration of Helsinki regarding consent and confidentiality. All patients signed an informed consent form according to the hospital model at the time of admission. The informed consent of the patients was fully respected in this study. The study received Ethics Council approval from the Neuropsychiatry Hospital, No. 8411/27 July 2023.

### 2.1. Study Design

Our retrospective study included 160 patients with ischemic and hemorrhagic stroke from a single center. They were divided into two equal groups, the study group (botulinum toxin group—BT) and the control group (electromyostimulation and radial shockwave group—ES), with 80 patients allocated to each group.

The patients included in the study were divided into two groups: those who underwent continuous hospitalization were included in the ES group, and those with daily hospitalization were included in the BT group. It is essential to note that, in accordance with the existing protocol in our country, the administration of botulinum toxin was restricted to patients undergoing daily hospitalization. The patient randomization was carried out based on the analysis of medical files that met the pre-defined inclusion criteria.

The retrospective study evaluated patients who had been hospitalized within the past three years. The data were initially collected during the patient’s first stage of treatment. Subsequently, a second stage of treatment occurred three months later, followed by a final evaluation during rehospitalization another three months later. The comparison between final evaluation data and the initial ones was then conducted.

The inclusion criteria were: age over 18 years, the presence of an ischemic or hemorrhagic stroke no earlier than 6 months, the absence of a recovery treatment or botulinum toxin injection (all patients were naive to this type of therapy), the presence spasticity of minimum grade 2 on Ashworth and Tardieu scales and readmission in hospital after 3 months with three distinct stages of assessment: initial, after three months, and final. Evaluations were conducted using the presented scales at both the initial and final stages. The exclusion criteria included patients who were not neurologically, pulmonarily, or cardiovascularly stable, patients with contraindications to electrical stimulation treatment or botulinum toxin injection, active infections, neurological diseases such as myasthenia gravis, presence of a pacemaker, cochlear implant, patients with skin lesions at the level of the skin that do not allow electrostimulation or injectable treatment with botulinum toxin, patients with oncological pathology, and those with severe depressive disorders, cachexia.

### 2.2. Evaluation Scales

The eligible patients for this study were evaluated for spasticity using the known scales. For our study, the selected scales were the modified Ashworth scale and the Tardieu scale. To assess functionality, we used the modified Frenchay scale. Muscle strength was evaluated by using the Medical Research Council (MRC) muscle strength scale, which ranges from 0 to 5. A score of 0 indicates lack of muscle tone, 1 indicates palpable muscle tone, 2 indicates movement in the horizontal plane, 3 indicates movement in the anti-gravitational plane but without resistance, 4 indicates movement in the anti-gravitational plane against slight resistance, and 5 indicates normal muscle strength [19].

The Barthel scale was used to explain the degree of incapacity and contains 10 items of activities rated with scores of 0, 5, 10, and 15, with only two items having 15 points, all the items being scored from 5 to 5 points, and a maximal total score of 100. The scores obtained represent the degree of dependence or independence as follows: 0–40 highly dependent, 45–60 partially dependent, 65–95 minimally dependent, 95–100 independent [20].

The modified Ashworth scale is a 4-point scale used to assess muscle spasticity. It ranges from 0 to 4, where 0 means no increase in muscle tonus, 1: slight increase in muscle tone with minimal resistance at the end of the movement, 1+: slight increase in muscle tonus with minimal resistance over half of the range of motion, 2: more marked increase in muscle tone with movement still possible, 3: considerable increase in muscle tone with passive movement difficult to achieve, 4: such an increase in muscle tonus (rigid) that the movement is impossible with the joint fixed in a flexed or extended position [21].

The Tardieu scale measures the reaction of the muscle group to stretch at a specific speed according to two parameters, as follows: the quality of the muscle reaction and the angle of muscle reaction. The speed is divided into 3 items from the slowest (V1) to the fastest (V3), and V2 is the speed at which the segment falls under gravity, and only V2 and V3 are used for evaluation. The quantified angles are represented by the R1 maximum speed catching angle and R2 at a slower speed. Evaluation was carried out at the same hour of the day and the tested limb must be placed in the same exact position in order to repeat the test [22]. The muscle reaction angle is measured in accordance with the minimum stretch position of the muscle [23]. The range is from 1 to 5, where 1 is no resistance on catching, 2 is a light resistance on catching at a precise angle and free catching at a precise angle, 3 the appearance of a non-persisting clonus within 10 s of precise angle catching, 4 is persistent clonus for more than 10 s at a precise angle, 5 is a fixed joint [24].

For a better accuracy of the functional evaluation, we use the modified Frenchay scale (MFS) that contains 10 activities, numbered from 1 to 10 with a total score of 100 for normal activity, moving from simple to complex activities. The activities must be completed by the patient using standardized everyday life instruments arranged in a semicircle shape at an arm’s length in front of the patient. The clinician asks the patient to execute six in two-handed tasks and four in one-handed tasks of the scale. The tasks are always carried out in the same order, from left to right; the one-handed tasks must be carried out with the impaired limb. Each task is evaluated from 0 (no movement) to 10 (normal movement). The patient is encouraged to practice stretching for the hyperactive muscles or spastic muscles of the tested upper limb between the two activities, in order to reduce transitory spastic dystonia and spastic contraction in order to be able to carry out the test [25]. The MFS is available on a public site together with some demonstrative videos [26].

Both groups received the same standardized kinetic therapy program, but instead of focal botulinum toxin injection the ES patients have followed a complex rehabilitation program including radial shockwave therapy for the spastic muscle groups and electrical stimulation with rectangular currents for the antagonist muscle groups to the spastic ones.

In order to achieve an adequate rehabilitation program, we need to maintain the spasticity at a reasonable level to allow joint mobility. Physical kinetic therapy is complex and consists in specific kinetic programs starting with stretching exercises and specific kinetic methods that improve the tonus of the paralyzed muscle groups and relieve the tonus in the spastic muscle groups. The kinetic therapy program must be learned and practiced at home in order to maintain the program and reduce spasticity. The foundation of kinetic programs is stretching elements applied at the spastic muscle groups as in Figure 1 for the hand and as in Figure 2 for the shoulder.

For the hands, the patient was trained and taught to perform specific stretching exercises for the flexors of the thumb, fingers, and wrist. The important muscles are flexor pollicis longus and brevis, opponents pollicis, adductor pollicis (Figure 1A), flexor digitorum profundus, flexor digitorum superficilis (Figure 1B), and flexor carpi radialis (Figura 1C). Also, the patients received a guide with exercises for other muscles that need to be stretched: brachiradialis, pronator teres, flexor carpi ulnaris.

Other types of exercises are for spastic muscle of the arm and shoulder (Figure 2). The patient was trained to perform specific postures that include stretching important spastic muscles: latissimus dorsi (Figure 2A), biceps (Figure 2B,C), and pectoralis (Figure 2D). The patients also received instruction for other muscles for the arm and shoulder: brachioradialis, pronator, teres, triceps brachii, deltoid (anterior and posterior parts).

The ES group received radial shockwave therapy on spastic muscle groups and electrical stimulation on antagonist groups (flabby muscle). The ES group received electrotherapy procedures like radial shockwave on spastic muscle groups with the purpose of fighting spasticity by lowering local fibrosis level and increasing the muscle elasticity [27,28]. The shockwave therapy used pulses at a pressure of 1.5 bars with a progressive increase to 2 bars; it started with 500 pulses, with 500 pulses every 3 days, up to a maximum of 1500 pulses per session, and the frequency used was 10 Hz. A program of 1 session every 2 days for 10 days was carried out, with a total of 5 applications per hospitalization period.

Also, the ES group received an electrical stimulation program using repetitive magnetic peripheral stimulation. This is necessary in order to stimulate the paralyzed muscle groups and restore the muscle balance with the spastic groups. Repetitive peripheral magnetic stimulation was preferred because the generated magnetic field will induce an electric stimulus that will produce a muscle contraction, a tetanic one, whereas the classical electrical stimulation produces a simple secusa type muscle contraction. The repetitive peripheral magnetic stimulation is thought to increase brain plasticity during muscle contraction by stimulating proprioceptive receptors [29]. The electrostimulation program used peripheral repetitive magnetic stimulation with a frequency of 30 HZ, a power of 30 W, and a stimulation rhythm of 2 s with a 6 s rest for 10 min per session for 10 days. It was applied to the paralyzed muscles to stimulate the movement and for visualization for the patients and the stretching of the spastic muscles during the stimulation. The ES group followed two stages of localized electrotherapy at a 3-month interval and received the same stretching exercises at home.

The study group (BT group) benefited from the effects of botulinum toxin, which has the role of reducing spasticity through its specific properties. Botulinum toxin—which is a protein neurotoxin produced by neurotoxigenic strains of anaerobic and spore-forming bacteria of the genus Clostridium—is commonly utilized and well tolerated [30] in neurorehabilitation as a treatment for focal spasticity [31] and reduces muscle hypertonia, improving functionality [32] and thus improving self-care activity by preventing contractures and their effects (vicious positions, immobilization) [18].

The BT group received a botulinum toxin injection treatment (incobotulinum toxin and abobotulinum toxin) in specific and permitted doses for the spasticity of the upper limb at 3-month intervals in three treatment stages. The doses were recommended and necessary according to the patient’s spasticity and did not exceed the maximum allowed doses.

Two types of toxins were used: incobotulinum toxin and abobotulinum toxin. A maximum of 1000 units of abobotulinum toxin and 200 U of incobotulinum toxin were used for the injection of the spastic muscles of the upper limb (100 U of incobotulinum toxin represents 400 U of abobotulinum toxin).

The patients received doses of a maximum of 1000 units of abobotulinum toxin injected in spastic muscles. For the round pronator, a maximum of 200 units was injected, for flexor digitorum profundus and superficial, a maximum of 400 units was used for each. For radial carpal flexor, a maximum of 150 units and for deltoid and pectoralis 250 U were used, for flexor pollicis longus a maximum of 100 units was injected. No more than 4 points were injected.

For incobotulinum toxin, a maximum of 200 units was injected. For the round pronator, a maximum of 50 units was injected, for flexor digitorum profundus and superficial, a maximum of 100 units was used for each. For radial carpal flexor, a maximum of 25 units was used, for deltoid and pectoralis 50 U was used, for flexor pollicis longus a maximum of 25 units was injected. No more than 4 points were injected.

The patients received a maximum of four echography-guided injections on the upper limb, including the shoulder muscles. The patients have received the same kinetic therapy program recommended at home which the patient had to execute every day. The two groups’ evaluations were initially made at 6 months, during which time both patient groups followed the kinetic program at home.

All patients from both groups received an exercise guide chosen for spastic muscles of the upper limb at discharge. Each stretching exercise must be held for 6–10 s with 20 s breaks. It is repeated 10 times in one session, 5 sessions per day.

### 2.3. Statistical Analysis

Descriptive statistics (mean ± standard deviation (SD) and median, interquartile range (IQR) for continuous variables or number and percentage for discrete variables) were used to assess the general characteristics of the patients. We performed a Kolmogorov–Smirnov test to check the distribution of the continuous variables for the population analyzed. Only the final Frenchay score in the study group presented a normal distribution and we used non-parametric tests to analyze these variables. The Wilcoxon signed-rank test was computed to determine the difference between pre-treatment (baseline) and post-treatment (follow-up) (botulinum toxin injection or electrostimulation). Spearman’s rho was used to analyze the correlation between improvement in spastic measurements and patients’ characteristics. GraphPad Prism 10.0.3 (GraphPad Software, Boston, MA, USA) was used for all analyses, with the statistical significance level set at *p* less than 0.05, two-tailed.

## 3. Results

One hundred and sixty patients were enrolled in this study, and baseline evaluations are described in Table 1. The average age of the participants was 63.3 ± 11.4, with younger patients in the group treated with botulinum toxin (*p*-value < 0.0001).

No differences were found between the two groups regarding the gender (59% males in BT group vs. 55% males in ES group, *p* = 0.75). Each group of patients included almost the same percentages of ischemic and hemorrhagic stroke (64% patients with ischemic stroke and 36% patients with hemorrhagic patients in BT group, respectively, 69% patients with ischemic stroke and 31% patients with hemorrhagic stroke in ES group, *p* = 0.616). The patients enrolled in the study presented different comorbidities, both in the BT group and in the ES group. These comorbidities were not contraindicated to the therapies applied for the two groups. Moreover, we wanted to observe the effectiveness of the therapy in the medical conditions presented by the patients. The main comorbidity was hypertension, with a relatively equal distribution in the two groups: 74 patients in the BT group and 78 in the ES group. Coronary heart disease (CHD) is the second most common comorbidity, with 65 patients in the BT group and 78 in the ES group. Atrial fibrillation (AF) and dyslipidemia (D) had a difference of about 10 patients between the two groups: 34 to 42 for AF and 66 to 73 for D. The distribution of diabetes mellitus (DM) was relatively equal, with 28 patients in the BT group and 24 in the ES group.

No significant differences were found between the two groups regarding the time since stroke, type of stroke, or the comorbidities (except for CHD: there were more patients with CHD in the second group, *p*-value = 0.001). One hundred and six patients had ischemic stroke, with no differences among patients from the two groups regarding the type of stroke (*p*-value = 0.616).

### 3.1. The Effect of BTx Injection and Electrostimulation on Upper Limb Spasticity

Compared to pre-injection evaluation, all the scores were significantly improved. Our analysis revealed that there is a statistically significant decrease between baseline and post-treatment for Ashworth and Tardieu scales in the case of the patients treated with BTx injection. The Ashworth scale was not significantly reduced in the post-intervention evaluation (*p*-value = 0.1745) for patients in the second group, as shown in Table 2. The MRC, Frenchay, and Barthel scores were significantly higher in the post-intervention evaluation for both groups.

### 3.2. Differences between the Improvements of the Analyzed Scales between the Two Treatments

Better improvements were obtained after the treatment with BTx injections than after electrostimulation, as in Figure 3. Regarding the spasticity scales, both Ashworth and Tardieu scales reported better improvements after the treatment with botulinum toxin than after electrosimulation. Thereby, the Ashworth scale (ASHW) improved with a mean ± SD of 1.86 ± 0.38 for the BT group and 0.15 ± 0.51 for the ES group (*p*-value < 0.0001). The Tardieu scale improved with a mean ± SD of 0.9 ± 0.49 for the BT group and 0.38 ± 0.49 for the ES group (*p*-value < 0.0001). The muscle strength measured with the MRC scale was more improved after botulinum toxin treatment than after electrosimulation: 1.71 ± 0.68 vs. 0.45 ± 0.65 (*p*-value < 0.0001). The degree of incapacity measured with the Barthel scale was also significantly improved after botulinum toxin treatment: 29.81 ± 6.44 vs. 6.94 ± 4.81 (*p*-value < 0.0001). The Frenchay scale was also significantly enhanced after the botulinum toxin treatment: 28.56 ± 9.63 vs. 4.18 ± 5 (*p*-value < 0.0001).

### 3.3. Correlations between the Measured Scales

The improvement in MRC was significantly related to age for both groups of patients, as shown in Figure 4. Age was an important prediction factor for better recovery in both groups, but not in all scales. In the BT group, better recovery was obtained for younger patients (for MRC scale: rho = −0.609, *p*-value < 0.0001; Tardieu scale: rho = −0.365, *p*-value = 0.001). For the ES group, better recovery was obtained for younger patients (for MRC scale: rho = −0.445, *p*-value < 0.0001; Barthel scale: rho = −0.239, *p*-value = 0.033; Tardieu scale: rho = −0.289, *p*-value = 0.009).

## 4. Discussion

The injection of botulinum toxin in the upper limb of a stroke patient must be administered correctly at well-established points to reach the spastic muscles after an objective assessment through precise scales. This is carried out by echo-guided injection to distribute the botulinum toxin only in the spastic muscles. This is one of the mandatory principles to be respected in order to obtain maximum results. Studies show that echo-guided botulinum toxin injection in stroke patients has superior results compared to unguided injection. The precise toxin injection into the target muscle is the key to an efficient and safe muscle spasticity treatment. Ultrasound-guided injection is recommended because, in the case of spastic muscle, the landmarks are modified compared to normal muscle [33]. The spastic muscle has a modified architecture determined by the loss of muscle mass and sarcomeres and the accumulation of connective tissue (fibrosis) and intramuscular fat, all leading to atrophy and shortening [34].

Spasticity must be combated objectively in order not to produce a muscular imbalance and allow physical therapy programs to be as effective as possible. The vast majority of studies have shown the effectiveness and safety of botulinum toxin treatment in patients with spasticity after stroke [35].

Our data regarding spasticity and functionality clearly demonstrated that the results were significantly better in the BT group, both in terms of improving spasticity and the degree of functionality measured by the Barthel and Frenchay scales. Moreover, muscle strength reaches an optimal level for functionality, and spasticity is reduced, leading to a decrease in disability. It is crucial to note that after BT injection, both muscle strength and functionality improve. This indicates that this therapy achieves a restoration of muscle balance between spastic flexors and weak extensors.

Adverse reactions were rare and consisted of pain at the injection site [36], flu-like symptoms, headache, nausea, and redness. Spasticity is recommended to be treated when it affects the life of the patient and his self-care capacity [37], and an early treatment may modify disease progression before secondary local biomechanical changes occur [16].

It has been proven in multiple studies that, after the injection, the patients were able to carry out active flexion/extension (of the fingers, of the fist, of the elbow) [38], improving the voluntary grip control in joints that were immobile before the injection. Also, better control of the antagonists was observed. The treatment with botulinum toxin had a beneficial effect in reducing spasticity [39] and the pain caused by it. Several authors and published studies highlight the role of the type of recovery used in patients undergoing treatment with botulinum toxin. The use of botulinum toxin in the treatment of post-stroke spasticity, followed by rehabilitation treatment such as kinetic therapy, radial shockwave, orthosis, neuromuscular electrostimulation etc., has been proven to improve motor control and coordination of the upper limb, this being highlighted by the modified Ashworth scale [40,41]. Thus, an increase in self-care capacity was observed, improvement in the use of the upper limb in patients with relatively preserved mobility, but also a reduction in pain and an increase in comfort for the patients with severe motor deficits [42].

Our study highlights, in addition to those presented, the importance of continuing the kinetic program during the period between botulinum toxin administrations, and thus the effect of the focal therapy is improved. The response to the treatment proved to be quite rapid, improving the patients’ lives. By reducing spasticity, it improves the patient’s mental state, thus increasing the chances that he will continue treatment with more involvement, a very important element in stroke recovery. The results of our study confirm the usefulness of using botulinum toxin in combination with the usual rehabilitation therapy in all patients in whom spasticity has become an impediment, regardless of the current stage of the disease evolution [36]. It has been clinically proven that treatment with botulinum toxin in patients with spasticity after stroke had beneficial effects in increasing the quality of life [43,44] by increasing self-care capacity, reducing pain, but also by reducing the involvement of family members and caregivers. The quality of life of stroke patients is also affected by the presence of spasticity. Botulinum toxin therapy has a favorable impact on the evolution of and increase in the quality of life of these patients by reducing spasticity [45]. Thus, if we are to talk only about the quality of the patient’s life, we must also take into account the fact that 60% of these patients also have a cognitive impairment [46]. Unlike other diseases with brain damage, such as primary brain tumors where the cognitive impairment is different and possibly dependent on the type of tumor [47], in stroke damage, cognitive decline is present in 60% of patients, to which if spasticity and motor deficit are added as sequelae in this category of patient, any therapy that can improve post-stroke status becomes important. Our study highlights the way to reduce spasticity through combined pharmacological and recovery therapies.

Current therapeutic strategies focus on new modalities of combined pharmacological and non-pharmacological treatment to obtain the best results. A systematic review highlights complex methods of medical intervention of ischemic stroke through a holistic approach that includes non-invasive, non-pharmacological therapeutic strategies [48].

Beyond the beneficial aspects regarding the improvement of the functionality of the use of botulinum toxin in the therapy of spasticity, it seems that the cost-effectiveness ratio is in favor of it, as a study from our country proves [49].

Related to the fact that this treatment is more efficient in the BT group, the results of our study show that younger patients have the ability to improve muscle reserve and reduce spasticity, and if we were to correlate with a study from Japan, it seems that elderly patients with stroke have a lower life expectancy compared to those without disabilities [50].

Our data also indicate that younger patients exhibit a superior muscle recovery capacity, whereas in older patients, the situation is more functional, with limited improvement in muscle strength. Younger patients not only show enhanced results on functional scales but also demonstrate predominant improvements in muscle tone and spasticity. These findings highlight that younger patients have a more favorable recovery outlook, benefiting from their structural and functional muscle reserve, whereas older patients primarily experience functional enhancements without significant muscle strength improvements [51].

There are studies that demonstrate that recovery must begin as early as possible, within days, starting from the onset of a stroke [52]. Our study draws attention to the situation, unfortunately not very rare, especially in countries with less access to medical rehabilitation services, that after a period of time starting from at least six months after the onset of a stroke, the results are beneficial, especially when the therapy is combined.

Spasticity is a syndrome with a variable clinical manifestation; thus, the dosage administered to each patient is adjusted based on the severity of spasticity observed. The prescribed doses are not standardized for individual patients; rather, they are determined on a per-spastic-muscle basis. Recommendations for botulinum toxin usage provide a range between minimum and maximum doses. Efforts are underway to establish a comprehensive guideline for botulinum toxin dosages specifically tailored to stroke patients [41,53].

In a 2020 review analyzing 25 studies involving 866 participants, shockwave therapy’s relationship with spasticity treatment was explored [27]. However, no studies have effectively compared groups undergoing current recovery treatments with shockwave and electrostimulation against those receiving botulinum toxin administration. Another review in 2021, comprising 33 studies and 1930 participants, emphasized the effectiveness of shockwave therapy in comparison to botulinum toxin injection. The authors noted the presence of several trials exploring the role of this application [54]. Notably, there is a lack of studies investigating the significance of continuing the stretching program in conjunction with botulinum toxin injection therapy or alternative recovery therapies. This association appears crucial for the success and sustained effectiveness of botulinum toxin therapy [54].

The post-stroke side-effects are multiple and, among them, the musculoskeletal ones are the most important. Spasticity results from the imbalance that occurs between inhibitory and facilitatory influences of descending pathways on the stretch reflex.

Unfortunately, at the present time, the genetic role of the development of spasticity after stroke is not well studied.

Synaptic plasticity is a factor that can influence spasticity after stroke (PSS). According to Le XIE et al., an agonist of the expression of some proteins such as brain-derived neurotrophic factor or growth-associated protein 43 that target synaptic plasticity at the cerebral synaptic level is beneficial from a therapeutic point of view in terms of post-stroke spasticity. So, those people that suffer a stroke with a genetically better synthesis of these proteins could delay the onset of or improve PSS [55].

Our study also evaluates the functionality of patients with spasticity in the upper limb. It is known that spasticity causes muscle dystonia with pain by affecting local biomechanics. Focal administration also improves muscle tone and implicitly decreases pain, which makes the functionality much improved after this treatment, as a study from 2022 tells us [56].

The limitations of our study are due to the fact that spasticity is not determined in the same way for each individual patient. Although spasticity is the same condition in all patients, its appearance differs from patient to patient, which is why the botulinum toxin doses cannot be the same for all patients. Another limitation of the study is the fact that spasticity is a syndrome, not a disease, which is constantly dynamic and can be negatively influenced by certain factors: low temperature, emotions, stress, etc., which will lead to an increase in spasticity. Similarly, other limitations are that electrostimulation programs and radial shockwave therapy parameters vary depending on the patient’s response to stimulation and spasticity presentation. This is why our study used two spasticity quantification scales. Another limitation could be the number of patients, and larger cohorts and more centers would be needed to obtain more data. This is also the reason why there are many discussions on the therapeutic approach in the medical literature. Stroke patient spasticity remains an element to be further studied both as a mechanism and as a method of treatment, using complex schemes and therapeutic associations that require the patient’s time and collaboration.

Building on this foundation, our study introduces a novel approach by compiling data related to spasticity and the level of independence achieved through combined therapy, comparing it to classical rehabilitation methods that incorporate modern electrostimulation techniques, radial shockwave therapy, and kinetic programs. While some benefits were also observed in the control group, focal therapy with botulinum toxin yielded superior results in improvement of functional capacity, as highlighted by the Barthel scale and MFS. Our study, carried out on relatively homogenous and numerically equal groups, systematically compared the applied programs and the resulting outcomes based on the therapies followed.

## 5. Conclusions

The complex approach to the patient with spasticity after stroke through focal therapy with botulinum toxin and the application of a well-defined stretching program improves both functional capacity and muscle strength, reducing spasticity and the degree of disability. Further trials are necessary to establish this association between botulinum toxin and a home training program for the successful maintenance of the therapy’s effectiveness.

## Figures and Tables

**Figure 1 life-13-02218-f001:**
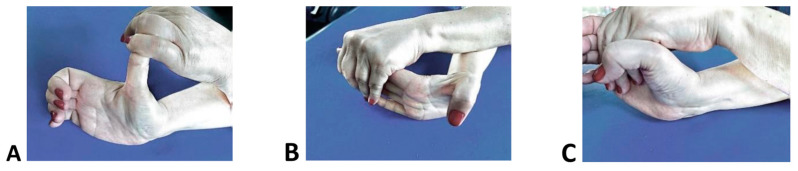
Hand stretching elements. (**A**). Stretching elements for flexor pollicis longus and brevis, opponens pollicis, adductor pollicis. (**B**). Stretching elements for flexor digitorum profundus, flexor digitorum superficilis. (**C**). Stretching elements for flexor carpi radialis.

**Figure 2 life-13-02218-f002:**
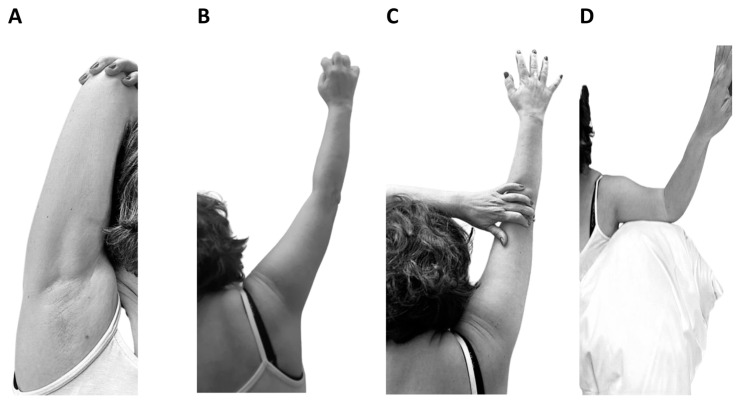
Stretching elements for arm and shoulder. (**A**). Stretching elements for latissimus dorsi. (**B**,**C**). Stretching elements for biceps. (**D**). Stretching elements for pectoralis.

**Figure 3 life-13-02218-f003:**
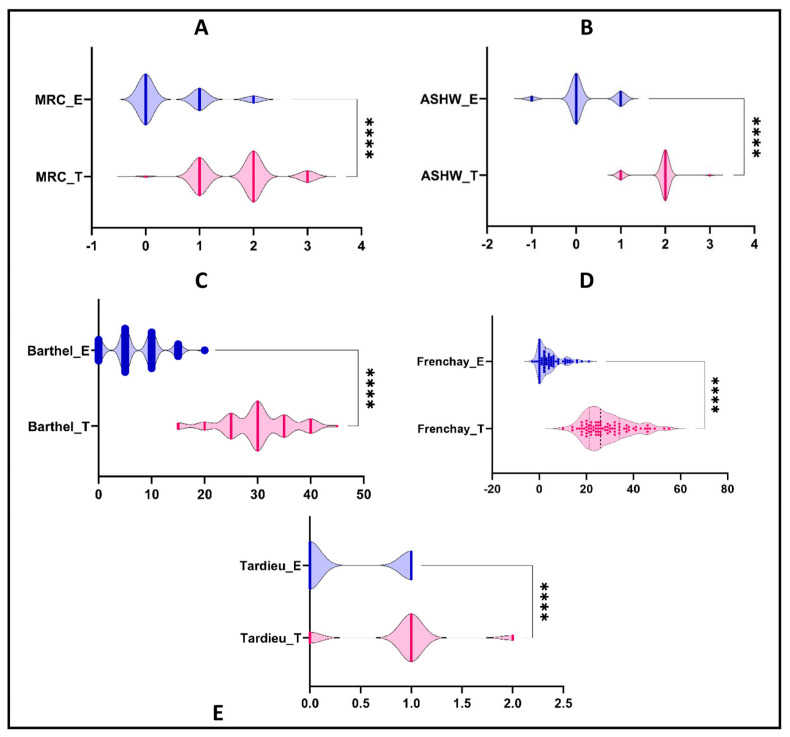
The differences between the two treatments’ enhancements. (**A**). Medical Research Council (MRC) scale. (**B**). Ashworth (ASHW) scale. (**C**). Barthel scale. (**D**). Frenchay scale. (**E**). Tardieu scale. T, BT group; E, Electrostimulation/ES group. ****, *p*-value < 0.0001.

**Figure 4 life-13-02218-f004:**
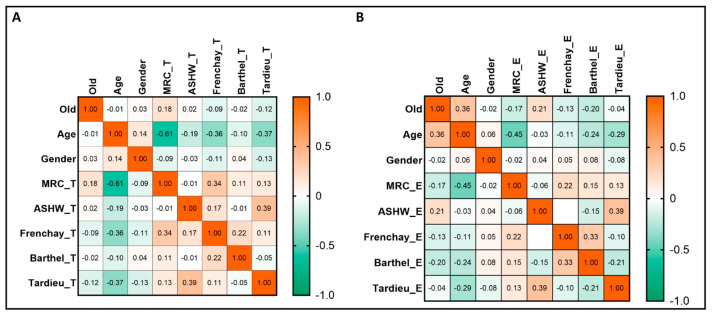
The correlations between the measured scales and patients’ characteristics. (**A**). The group of patients treated with botulinum toxin injection. (**B**). The group of patients treated with electrostimulation. The heatmaps visually represent the Spearman’s rho coefficients ranging from orange for positive correlation and green for negative correlation.

**Table 1 life-13-02218-t001:** Baseline patient characteristics.

Characteristics	Botulinum Toxin Group (BT Group) (n = 80)	Electrostimulation and Shockwave Group (ES Group) (n = 80)	*p*-Value
Age, years	58.74 ± 10.56 61.5 (54.25–66)	67.76 ± 10.51 67 (62.3–74.5)	<0.0001
Gender, male	47 (58.8%)	44 (55.0%)	0.750
Time since stroke	3.52 ± 2.09 3 (2–5)	3.39 ± 2.92 2 (1–5)	0.124
Stroke type Ischemic Hemorrhagic	51 (63.8%) 29 (36.2%)	55 (68.8%) 25 (31.2%)	0.616
Hypertension, yes	74 (92.5%)	78 (97.5%)	0.276
Coronary heart disease, yes	65 (81.3%)	78 (97.5%)	0.001 ***
Atrial fibrillation, yes	34 (42.5%)	42 (52.5%)	0.268
Diabetes, yes	28 (35.0%)	24 (30.0%)	0.613
Dyslipidemia, yes	66 (82.5%)	73 (91.3%)	0.159

***, *p*-value < 0.001.

**Table 2 life-13-02218-t002:** Baseline scales comparison vs. post-treatment for the two treatment groups.

Characteristics Mean ± SD Median (IQR)	Botulinum Toxin Group (BT Group) (n = 80)			Electrostimulation and Shockwave Group (ES Group) (n = 80)		
	Baseline	Post-Treatment	*p*-Value	Baseline	Post-Treatment	*p*-Value
Medical Research Council scale	1.6 ± 0.5 2 (1–2)	3.3 ± 0.7 3 (3–4)	<0.0001 ****	1.8 ± 0.6 2 (1–2)	2.2 ± 0.7 2 (2–3)	<0.0001 ****
Ashworth	3.1 ± 0.2 3 (3–3)	1.2 ± 0.4 1 (1–1)	<0.0001 ****	2.5 ± 0.8 3 (2–3)	2.4 ± 0.6 2 (2–3)	0.1745
Frenchay	19.4 ± 9.3 20 (11–28.8)	47.9 ± 7.7 48 (42.3–53)	<0.0001 ****	24.6 ± 10.8 25 (13.5–32)	28.8 ± 11.3 30 (22–34)	0.0180 *
Tardieu	3.4 ± 0.6 3 (3–4)	2.5 ± 0.5 2 (2–3)	<0.0001 ****	3.1 ± 0.7 3 (3–3)	2.8 ± 0.6 3 (2–3)	0.0006 ***
Barthel	25 ± 6.9 25 (20–30)	54.8 ± 7.3 55 (50–60)	<0.0001 ****	26.9 ± 10.6 25 (20–35)	33.9 ± 11.2 35 (25–40)	0.0001 ***

SD, standard deviation; IQR, interquartile range; *, *p*-value < 0.05; ***, *p*-value < 0.001; ****, *p*-value < 0.0001.

## Data Availability

The research data are available from the correspondence author upon request.

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
