# Peer review of "Management of Upper-Limb Spasticity Using Modern Rehabilitation Techniques versus Botulinum Toxin Injections Following Stroke"

_life, 2023, doi:10.3390/life13112218_

Round 1
Reviewer 1 Report
Comments and Suggestions for Authors
Dear Author,
I appreciate the effort and depth of research presented in your manuscript on treating post-stroke spasticity using botulinum toxin. Here are eight detailed recommendations to improve the clarity and depth of your study:
Introduction Clarity: Elaborate more on the background of using botulinum toxin in medical treatments. It will provide readers with a clearer context.
Study Sample: Specify the number of participants in your study, their demographics, and the criteria for selection. This will enhance the validity of the results.
Dosage Variability: The manuscript mentions that botulinum toxin doses cannot be the same for all patients. Provide more explicit details or examples of the variability to help readers understand the range and reasons behind these differences.
Methodological Consistency: While you mentioned various treatments like electro stimulation and radial shockwave therapy, more details on the frequency, duration, and intensity of these treatments for the control and test groups would improve clarity.
Comparison with Previous Studies: Elaborate more on how your findings compare with or deviate from previously published studies. This could strengthen the uniqueness or importance of your study.
Conclusion Elaboration: While your conclusions are precise, consider elaborating on the potential implications of your findings for future research or clinical practices.
Sample Size and Demographics: While the content mentions "relatively homogenous and numerically equal batches," the actual number of participants, their age, gender, and other relevant demographics aren't provided.
Randomization and Control: It’s mentioned there is a control group, but there's no explicit mention of how participants were randomized into either the treatment or control group. The nature and exact treatment of the control group are also not detailed.
Procedure Details: Although various treatments are listed (e.g., botulinum toxin administration, kinetic therapy, radial shockwave, etc.), the exact procedure for each treatment, their frequency, duration, and dosage (especially for botulinum toxin), isn't provided.
Measurement Tools: There's mention of "two spasticity quantification scales" and the "Barthel and MFS scales," but the paper does not detail how these scales were used, their validity and reliability, or the exact metrics they provided.
Data Analysis: There's no information on how the data were analyzed. Were there any statistical tests used? If so, which ones? What significance levels were set?
Duration of the Study: Over what period was this study conducted? Was it over weeks, months, or years?
Thank you for your contribution to the field. I believe that with these adjustments, your manuscript will offer a more comprehensive and clearer understanding of the topic.
Author Response
We extend our sincere gratitude for the constructive comments provided by you. Your insightful feedback has been invaluable in refining our manuscript. We also appreciate the time and effort you dedicated to reviewing our work thoroughly.
In response to your comments, we have diligently addressed each point raised, implementing necessary corrections in our manuscript. These revisions have been meticulously made using the tracked changes feature, ensuring transparency and traceability in the editing process. Your feedback has significantly contributed to the improvement of the quality and clarity of our research, and we are grateful for your input.
Q 1
Introduction Clarity: Elaborate more on the background of using botulinum toxin in medical treatments. It will provide readers with a clearer context.
Thank you for your observation. After carefully reading the paragraphs, we modified the abstract as you suggested to be more accurate and clearer as follows:
Our purpose is to emphasize the role of botulinum toxin in spasticity therapy and functional recovery in patients with stroke. Our retrospective study compared two groups with ischemic and hemorrhagic stroke. The study group (BT group) included 80 patients receiving focal botulinum toxin as therapy for the upper limb in spastic muscle at every three months, for three times. The control group (ES group) included 80 patients who received only medical rehabilitation consisting in electro-stimulation and radial shockwave therapy for the upper limb applied every three months for three times. Both groups received the same stretching program for spastic muscles as a home training program. We evaluated the evolution of the patients using specific scales: muscle strength, Ashworth, Tardieu, Frenchay, and Barthel. The analysis indicated a statistically significant difference between the two groups of all scales analyzed, with better results for the BT group (p<0.0001 for all scales). In our study, the age of the disease was an important prediction factor for better recovery in both groups, but not in all scales. Better recovery was obtained for younger patients (in BT group, MRC scale: rho=-0.609, p-value<0.0001; Tardieu scale: rho=-0.365, p-value=0.001; in Control group, MRC scale: rho=-0.445, p-value<0.0001; Barthel scale: rho=-0.239, p-value=0.033). Our results demonstrated the effectiveness of botulinum toxin therapy compared with rehabilitation method with the reduction of the recovery time of the upper limb, as well as the improvement of functionality and reduction of disability. Although all patients followed a specific kinetic program, important improvements were evident in the botulinum toxin group.
Q 2
Study Sample: Specify the number of participants in your study, their demographics, and the criteria for selection. This will enhance the validity of the results.
We detailed the inclusion and exclusion criteria of the patients in our study in the section Study design from Materials and Methods, as below:
Our retrospective study included 160 patients with ischemic and hemorrhagic stroke from a single center. They were divided into two equal groups, the study group (Botulinum toxin group - BT) and the control group (electromyostimulation and radial shockwave group - ES), with 80 patients allocated to each group.
The patients included in the study were divided into two groups: those who underwent continuous hospitalization were included in the ES group, and those with daily hospitalization were included in the BT group. It is essential to note that, in accordance with the existing protocol in our country, the administration of botulinum toxin was restricted to patients undergoing daily hospitalization. The randomization of patients was carried out based on the analysis of medical files that met the predefined inclusion criteria.
The retrospective study evaluated patients who had been hospitalized within the past three years. The data were initially collected during the patient’s first stage of treatment. Subsequently, a second stage of treatment occurred three months later, followed by a final evaluation during re-hospitalization another three months later. The comparison between final evaluation data and the initial ones was then conducted.
The inclusion criteria were age over 18 years, the presence of an ischemic or hemorrhagic stroke no earlier than 6 months, the absence of a recovery treatment or botulinum toxin injection (all patients were naive to this type of therapy), the presence spasticity of minimum grade 2 on Ashworth and Tardieu scale and readmission in hospital after 3 months with three distinct stages of assessment: initially, after three months, and finally. Evaluations were conducted using the presented scales at both the initial and final stages.. The exclusion criteria were patients who were not neurologically, pulmonary or cardiovascular stable, patients with contraindications to electrical-stimulation treatment or botulinum toxin injection, active infections, neurological diseases such as myasthenia gravis, presence of a pacemaker, cochlear implant, patients with skin lesions at the level of the skin that do not allow electrostimulation or injectable treatment with botulinum toxin, patients with oncological pathology and those with severe depressive disorders, cachexia.
Q 3
Dosage Variability: The manuscript mentions that botulinum toxin doses cannot be the same for all patients. Provide more explicit details or examples of the variability to help readers understand the range and reasons behind these differences.
We described the doses on Materials and Methods section
Two distinct types of toxins were employed in the study: incobotulinum toxin and abobotulinum toxin. The injection of spastic muscles in the upper limb involved a maximum of 1000 units of abobotulinum toxin and, correspondingly, 200 units of incobotulinum toxin (correlating to the ratio of 100 units of incobotulinum toxin representing 400 units of abobotulinum toxin). The patients received doses of a maximum of 1000 Abobotulinum toxin distributed on spastic muscles injected. For the round pronator maximum 200 units were injected, and for the flexor digitorum a maximum profundus and superficial maximum of 400 units each. For radial carpal flexor maximum of 150 units and for deltoid and pectoralis 250 U, for flexor pollicis longus maximum 100 units were injected. No more than 4 points were injected.
For incobotulinum toxin, a maximum of 200 units were injected. For the round pronator maximum 50 units were injected, for flexor digitorum a maximum profundus and superficial maximum of 100 units for each. For radial carpal flexor maximum of 25 units, for deltoid and pectoralis 50 U, for flexor pollicis longus maximum of 25 units were injected. No more than 4 points were injected.
We introduced a new discussion about the doses with a new reference as below:
Spasticity is a syndrome with a variable clinical manifestation; thus, the dosage administrated to each patient is adjusted based on the severity of spasticity observed. The prescribed doses are not standardized for individual patients; rather, they are determined on a per-spastic-muscle basis. Recommendations for botulinum toxin usage provide a range between minimum and maximum doses. Efforts are underway to establish a comprehensive guideline for botulinum toxin dosages specifically tailored to stroke patients (41, 53)
- Doan, T.N.; Kuo, M. Y.; Li-Wei Chou, L. W; Efficacy and Optimal Dose of Botulinum Toxin A in Post-Stroke Lower Extremity Spasticity: A Systematic Review and Meta-Analysis, Toxins (Basel). 2021. doi: 10.3390/toxins13060428
Q4
Methodological Consistency: While you mentioned various treatments like electro stimulation and radial shockwave therapy, more details on the frequency, duration, and intensity of these treatments for the control and test groups would improve clarity.
Thank you for your observation. We agree and more details were included in our manuscript:
The electro-stimulation program used peripheral repetitive magnetic stimulation with a frequency of 30 HZ, a power of 30 W, and a stimulation rhythm of 2 seconds with a 6-second rest for 10 minutes per session, 10 days. It was applied to the paralyzed muscles to stimulate the movement, the visualization to the patients and the stretching of the spastic muscles during the stimulation.
The shockwave therapy used pulses at a pressure of 1.5 bars with a progressive increase to 2 bars, it started with 500 pulses, with 500 pulses every 3 days, up to a maximum of 1500 pulses per session, and the frequency used was 10 Hz. A program of one session every two day for 10 days was carried out, with a total of 5 applications per hospitalization period.
Q 5
Comparison with Previous Studies: Elaborate more on how your findings compare with or deviate from previously published studies. This could strengthen the uniqueness or importance of your study.
We added more comparisons with previous published studies and included more references:
In a 2020 review analysing 25 studies involving 866 participants, shock wave therapy's relationship with spasticity treatment was explored [27]. However, no studies have effectively compared groups undergoing current recovery treatments with shockwave and electro-stimulation against those receiving botulinum toxin administration. Another review in 2021, comprising 33 studies and 1930 participants, emphasized the effectiveness of shock wave therapy in comparison to botulinum toxin injection. The authors noted the presence of several trials exploring the role of this application [54]. Notably, there is a lack of studies investigating the significance of continuing the stretching program in conjunction with botulinum toxin injection therapy or alternative recovery therapies. This association appears crucial for the success and sustained effectiveness of botulinum toxin therapy [54].
Q 6
Conclusion Elaboration: While your conclusions are precise, consider elaborating on the potential implications of your findings for future research or clinical practices.
Conclusions
The complex approach of the patient with spasticity after stroke through focal therapy with botulinum toxin and the application of a well-defined stretching program improve both functional capacity and muscle strength, reducing spasticity and the degree of disability. Our research could be the start of other studies for this specific field of spasticity. Further trials are necessary for establishing this association between botulinum toxin and a home training program for the successful maintenance of the therapy’s effectiveness.
Q 7
Sample Size and Demographics: While the content mentions "relatively homogenous and numerically equal batches," the actual number of participants, their age, gender, and other relevant demographics aren't provided.
These aspects are already specified in the question 2.
Q 8
Randomization and Control: It’s mentioned there is a control group, but there's no explicit mention of how participants were randomized into either the treatment or control group. The nature and exact treatment of the control group are also not detailed.
We included these details about our study design in our answer to Q 2 and we explained the therapy in the Materials and Methods section.
Q 9
Procedure Details: Although various treatments are listed (e.g., botulinum toxin administration, kinetic therapy, radial shockwave, etc.), the exact procedure for each treatment, their frequency, duration, and dosage (especially for botulinum toxin), isn't provided.
We explained more of these details on Materials and methods section.
Q 10
Measurement Tools: There's mention of "two spasticity quantification scales" and the "Barthel and MFS scales," but the paper does not detail how these scales were used, their validity and reliability, or the exact metrics they provided.
We mentioned this on the Evaluation scales as below:
The Ashworth and Tardieu scales are scales for spasticity scorification, and the Barthel scale is a scale of incapacity, useful in evaluating patients dynamically after the applied treatment. The Franchay scale is a scale of functionality, useful in the dynamic assessment of patients.
Q 11
Data Analysis: There's no information on how the data were analyzed. Were there any statistical tests used? If so, which ones? What significance levels were set?
The information on how the data were analyzed is presented in the subsection Statistical analyses from Materials and Methods. The Wilcoxon signed-rank test was computed to determine the difference between pre- (baseline) and post- (follow-up) treatment (botulinum toxin injection or electro-stimulation). The statistical significance level was set at p less than 0.05, two-tailed.
Q 12
Duration of the Study: Over what period was this study conducted? Was it over weeks, months, or years?
We presented all this data on study design as you recommended. Thank you.
The retrospective study evaluated patients who had been hospitalized within the past three years. The data were initially collected during the patient’s first stage of treatment. Subsequently, a second stage of treatment occurred three months later, followed by a final evaluation during re-hospitalization another three months later. The comparison between the final evaluation data and the initial ones was then conducted.
Reviewer 2 Report
Comments and Suggestions for Authors
Authors presented a retrospective comparative study of the upper limb spasticity management in patients with stroke. The study design compared two homogeneus group: the first group (study group) included patients with focal botulinum toxin as therapy for the upper limb; the second group (Control group) received only electro-stimulation and radial shockwave therapy for the upper limb. Both groups received the same stretching program for spastic muscles.
The fingings of the proposed study suggest the effectiveness of botulinum toxin therapy and the reduction of the recovery time of the upper limb, the improvement of muscle strenght of function and of quality of life.
I strongly suggest to consider the role of genetic factor in stroke (10.1016/j.gene.2022.146880 / 10.1007/s00439-012-1224-9 / 10.19852/j.cnki.jtcm.20221108.002 ) especially for spasticity; and the role of management of pain (10.3390/toxins14010039 /10.3390/jcm10235552 /10.1161/STROKEAHA.111.671008 ).
Author Response
The authors express their sincere gratitude to the reviewer for their thoughtful feedback, kind words, and valuable suggestions, all of which significantly contributed to enhancing the quality of our work. The meticulous revisions and refinements recommended by the reviewer were incorporated into the paper using the "Track Changes" feature.
Q 1
strongly suggest to consider the role of genetic factor in stroke (10.1016/j.gene.2022.146880 / 10.1007/s00439-012-1224-9 / 10.19852/j.cnki.jtcm.20221108.002 ) especially for spasticity
We added more discussion and new references to our manuscript. Thank you for your suggestion.
The post-stroke side-effects are multiple, among them the musculoskeletal ones are the most important. Spasticity results from the imbalance that occurs between inhibitory and facilitatory influences of descending pathways on the stretch reflex.
Unfortunately, at the present time, the genetic role of the development of spasticity after stroke is not well studied.
Synaptic plasticity is a factor that can influence spasticity after stroke (PSS). According to Le XIE et all, an agonist of the expression of some proteins as brain-derived neurotrophic factor or growth associated protein 43 that target synaptic plasticity at the cerebral synaptic level brings a benefit from a therapeutic point of view in terms of post-stroke spasticity. So those people who suffer a stroke with a genetically better synthesis of these proteins could delay the onset or improve PSS [55].
Q 2
the role of management of pain (10.3390/toxins14010039 /10.3390/jcm10235552 /10.1161/STROKEAHA.111.671008 ).
We added more discussion and new references to our manuscript. Thank you for your suggestion.
Our study also evaluates the functionality of patients with spasticity in the upper limb. It is known that spasticity causes muscle dystonia with pain, by affecting local biomechanics. Focal administration also improves muscle tone and implicitly decreases pain, which makes the functionality much improved after this treatment, as a study from 2022 tells us [56].
Reviewer 3 Report
Comments and Suggestions for Authors
This is a retrospective study that compared the efficacy of botulinum toxin injection (study group) versus electro-stimulation and radial shockwave therapy (control group) on spasticity, strength, and function in stroke patients. Data were gathered from a single center. I have several concerns that are presented below.
1- The title is generic, I suggest that the authors modify it and include a title that refers to the study interventions and main result.
2- The study has serious problems that have an impact on the findings and conclusion. The study examined the differences between the effects of injections of botulinum toxin and shockwave therapy and electrostimulation. It doesn't seem workable to compile information on how shockwave therapy and electrostimulation affect spasticity in this comparison. It would have been preferable if each had just been contrasted with Botox injections; that is, the research ought to have looked at three groups (Botox vs shockwave vs electrostimulation).
3- The objective of the study is not clear. Authors need to develop a study objective that effectively communicates the purpose, scope, and significance of the study, guides the study design, and provides a clear direction for the investigation.
4- I could not follow the results and discussion based on the stated objective. The study objective of a study sets the foundation for the research and serves as a guide for the study design, data collection, and analysis.
5- The rationale of the study is not entirely evident. The authors may be required to furnish additional context and background details regarding the role of the evaluated interventions for stroke patients, as well as to emphasize the reasoning based on the extant literature.
6- The same stretching regimen for spastic muscles was given to both groups. Was the regimen for physical rehabilitation limited to stretching alone? This is crucial since it could skew the findings.
7- You used particular scales to assess the patient's progress. What was the time frame of the assessment? The temporal sequence between exposures and outcomes should be established accurately.
8- This was a retrospective study. Retrospective studies rely on existing data from medical records, databases, or registries. Researchers have no control over the quality or completeness of the data collected, which can introduce biases and confounding factors. Missing or inaccurate data can undermine the study's integrity. I found myself inquisitive to understand how the authors controlled for the outcome assessment and quality of the data that they collected from the medical records.
9- Also, retrospective studies often involve selecting participants based on their medical records, which may not accurately represent the target population. This selection bias can introduce systematic differences between the study group and the general population, affecting the generalizability of the results.
10- Moreover, it may struggle to account for confounding variables, which are factors that can influence both the exposure and outcome being studied. Since the data is collected after the events have occurred. Did the authors have access to all relevant information or were able to control for confounders adequately? This can lead to misleading associations or erroneous conclusions.
11- How was the level of spasticity assessed in the study participants? Was this in the same way?
12- Should the data from a single center be enough to provide transferrable evidence about the role of these interventions? Would the results be different if data were collected from different centers in different geographic areas?
13- One requirement for inclusion was a six-month period following the stroke. Have you considered the time elapsed since the stroke as a covariate in the analysis?
14- Patients in the study group received a maximum of four echography-guided injection sites on the upper limb, including the shoulder muscles. What about the injection doses?
15- The study is missing details about the treatment parameters for shockwave and electrostimulation.
16- Material and methods should be re-organized. I suggest creating subtitles to improve readability.
17- The authors gave too many unnecessary details for the assessment and interventions. You should focus on explicitly discussing the procedures.
18- Although data from 160 patients seems to be enough, I am not sure if the study is powered enough for the study design and outcome measures.
19- Tables:- why did you label the control group as an electrostimulation group although the participants of this group also received shockwave?
20- The discussion could have been better developed. The authors should have focused on interpreting and discussing the study's findings, comparing them with previous research, and exploring their implications.
21- A large space should have also been spared to highlight the study limitations and indicate how cautious readers should be in interpreting these findings.
Comments on the Quality of English LanguageModerate editing of the English language required to improve readibility
Author Response
The authors express their gratitude for the insightful and stimulating comments and suggestions provided. Additionally, we extend our thanks for the careful observation of errors and the effort taken to bring them to our attention, allowing us the opportunity to clarify and rectify the paper accordingly. As a general approach to enhance the clarity and precision of the manuscript, changes were made to its content using the "Track Changes" feature. These revisions were undertaken with a commitment to improving the overall coherence and comprehensibility of the research presented in the paper. We appreciate the valuable input, which has played a pivotal role in refining the quality of our work.
Q 1
The title is generic, I suggest that the authors modify it and include a title that refers to the study interventions and main result.
We admit your idea is better than ours, and we changed the title as below:
Management of upper limb spasticity using modern rehabilitation techniques versus botulinum toxin injection
Q 2
The study has serious problems that have an impact on the findings and conclusion. The study examined the differences between the effects of injections of botulinum toxin and shockwave therapy and electrostimulation. It doesn't seem workable to compile information on how shockwave therapy and electrostimulation affect spasticity in this comparison. It would have been preferable if each had just been contrasted with Botox injections; that is, the research ought to have looked at three groups (Botox vs shockwave vs electrostimulation).
We sincerely appreciate your feedback and have duly incorporated the suggested changes into the text. To address your recommendation, we have added more details, ensuring accuracy and adherence to the specified format. Thank you for your valuable input, which has contributed to the precision and clarity of our presentation.
The main objective of the study is to compare the effectiveness of botulinum toxin versus modern spasticity and functionality rehabilitation techniques. This comparison includes the association of electromyostimulation, which, by contracting paralyzed muscles, initiates movement in the spastic upper limb. Shockwave therapy has the role of reducing spasticity by targeting the spastic muscles. Thus, the association of the two therapies impacts both spasticity and functionality. Botulinum toxin therapy, by reducing spasticity, allows for much easier movements and improves functionality. The kinetic stretching program applied to both groups has a role in lengthening the spastic hamstrings and indirectly in improving functionality. Thus, spasticity and functionality are controlled by two methods: pharmacological and rehabilitation techniques.
If an electrostimulation group were also chosen, the results would not be objective because electrostimulation as the only therapy has no role in spasticity; it helps the functionality, the increase in muscle strength, which the study quantifies through the Barthel, Franchay and MRC scales.
Q 3
The objective of the study is not clear. Authors need to develop a study objective that effectively communicates the purpose, scope, and significance of the study, guides the study design, and provides a clear direction for the investigation.
We reformulated the objectives as below:
The objectives of our study are to highlight the differences between botulinum toxin type A therapy (incobotulinum toxin or abobotulinum toxin) and recovery therapy, combined with specific kinetic programs, in the management of spasticity and functionality in stroke patients.
We presented a detailed study design (lines 163-216).
Q 4
I could not follow the results and discussion based on the stated objective. The study objective of a study sets the foundation for the research and serves as a guide for the study design, data collection, and analysis.
We revised the study objectives and the data collection method in the research design.
The discussions are related to the results obtained on the spasticity and functionality scales for both groups. More than that, it was pointed out that in the literature the improvement of functionality led to an increase in the quality of life and cognitive aspects, compared to other pathologies where spasticity indirectly occurs (stroke, tumors).
Q 5
The rationale of the study is not entirely evident. The authors may be required to furnish additional context and background details regarding the role of the evaluated interventions for stroke patients, as well as to emphasize the reasoning based on the extant literature.
We modified the abstract and added a sentence with a clear presentation of the study's objectives. Also, we competed the discussion with more references.
Q 6
The same stretching regimen for spastic muscles was given to both groups. Was the regimen for physical rehabilitation limited to stretching alone? This is crucial since it could skew the findings.
The stretching program was the same for both groups, like a home training program after discharge. It is a department protocol, all patients with spasticity receive an exercise guide to continue at home.
Q 7
You used particular scales to assess the patient's progress. What was the time frame of the assessment? The temporal sequence between exposures and outcomes should be established accurately.
We presented all this data on study design. The retrospective study evaluated hospitalized patients in the last 3 years. The data were taken initially when the patient presented for the first stage of treatment, then three months later the second stage of treatment followed, and after another three months at re-hospitalization, the final evaluation data were taken and the comparison with the initial ones was made.
Q 8
This was a retrospective study. Retrospective studies rely on existing data from medical records, databases, or registries. Researchers have no control over the quality or completeness of the data collected, which can introduce biases and confounding factors. Missing or inaccurate data can undermine the study's integrity. I found myself inquisitive to understand how the authors controlled for the outcome assessment and quality of the data that they collected from the medical records.
The collected data were selected from the existing scales in the medical files, being a unique protocol and these data can be found, so there is no risk that the information is not complete. Were the same examiners who used the scales presented for the spasticity patients in the study.
Q 9
Also, retrospective studies often involve selecting participants based on their medical records, which may not accurately represent the target population. This selection bias can introduce systematic differences between the study group and the general population, affecting the generalizability of the results.
Only the cases that met the inclusion criteria were selected in the presented study.
Q 10
Moreover, it may struggle to account for confounding variables, which are factors that can influence both the exposure and outcome being studied. Since the data is collected after the events have occurred. Did the authors have access to all relevant information or were able to control for confounders adequately? This can lead to misleading associations or erroneous conclusions.
For the presentation of the study, only the scales found in the medical files and those that met the inclusion and exclusion criteria were collected. Situations with incomplete files were excluded.
Q 11
How was the level of spasticity assessed in the study participants? Was this in the same way?
All patients were evaluated for spasticity using two scales Ashworth and Tardieu precisely for accuracy of spasticity evaluation.
Q 12
Should the data from a single center be enough to provide transferrable evidence about the role of these interventions? Would the results be different if data were collected from different centers in different geographic areas?
The study was single center, being the only neurological recovery clinic in our region. Moreover, there are neurology department that carry out treatment with botulinum toxin, but they do not have the possibility to carry out a recovery program. The other recovery centers do not use botulinum toxin in the treatment of spasticity, and patients are sent to our clinic. Probably if there were other centers specifically for treatment with botulinum toxin and recovery, we could make more comparisons, but at this moment, our clinic is the only one in the region with a large cohort of patients from which we could select the participants for our study.
Q 13
One requirement for inclusion was a six-month period following the stroke. Have you considered the time elapsed since the stroke as a covariate in the analysis?
This six-month interval was chosen because spasticity requires time to settle, which time is variable from patient to patient. Otherwise, if we do not have a well-established degree of spasticity, for botulinum toxin treatment, we do not have accurate results. For a fair comparison, patients with a minimum age of 6 months and a degree of spasticity well specified in the inclusion criteria were chosen.
Q 14
Patients in the study group received a maximum of four echography-guided injection sites on the upper limb, including the shoulder muscles. What about the injection doses?
We completed this information in the Materials and methods section.
The patients received doses of a maximum of 1000 Abobotulinum toxin distributed on spastic muscles injected. For the round pronator maximum 200 units were injected, for flexor digitorum profundus and superficial maximum 400 units for each. For radial carpal flexor maximum 150 units and for deltoid and pectoralis 250 U, for flexor pollicis longus maximum 100 units were injected. No more than 4 points were injected.
For incobotulinum toxin, a maximum of 200 units were injected. For the round pronator maximum 50 units were injected, for flexor digitorum a maximum profundus and superficial maximum of 100 units for each. For radial carpal flexor maximum 25 units, for deltoid and paectoralis 50 U, for flexor pollicis longus maximum 25 units were injected. No more than 4 points were injected.
These 4 injection points represent the maximum number of points recommended per session by the manufacturer and the department's protocol. There is no guide, but the recommendations are also to focally inject the spastic muscle, not to spread the amount of substance in several areas, but to go strictly focally, in the target muscle.
Q 15
The study is missing details about the treatment parameters for shockwave and electrostimulation.
We completed this information in the Materials and methods section.
The electro-stimulation program used peripheral repetitive magnetic stimulation with a frequency of 30 HZ, a power of 30 W, and a stimulation rhythm of 2 seconds with a 6-second rest for 10 minutes per session, 10 days. It was applied to the paralyzed muscles to stimulate the movement, the visualization to the patients and the stretching of the spastic muscles during the stimulation.
The shockwave therapy used pulses at a pressure of 1.5 bars with a progressive increase to 2 bars, it started with 500 pulses, with 500 pulses every 3 days, up to a maximum of 1500 pulses per session, and the frequency used was 10 Hz. A program of one session every two day for 10 days was carried out, with a total of 5 applications per hospitalization period.
Q 16
Material and methods should be re-organized. I suggest creating subtitles to improve readability.
We reorganized the article adding subtitles to improve readability: Study design, Evaluation scales, Statistical analysis.
Q 17
The authors gave too many unnecessary details for the assessment and interventions. You should focus on explicitly discussing the procedures.
I have reorganized the article and presented separately each evaluation and its role.
Q 18
Although data from 160 patients seems to be enough, I am not sure if the study is powered enough for the study design and outcome measures.
After a post hoc power computation (G*Power 3.1.9.7, University of Dusseldorf, Germany), given α=0.05, sample size = 80, we obtained the power of the test equal to 1.00, as in the below figures.
Q 19
Tables:- why did you label the control group as an electrostimulation group although the participants of this group also received shockwave?
Thank you for highlighting this matter, we changed the control group’s label in ES – abbreviating electrostimulation and shockwave treatment.
Q 20
The discussion could have been better developed. The authors should have focused on interpreting and discussing the study's findings, comparing them with previous research, and exploring their implications.
New citations were introduced for the requested additional comparisons, presented in Discussions.
Q 21
A large space should have also been spared to highlight the study limitations and indicate how cautious readers should be in interpreting these findings.
We modified the discussion related with the limitations of our study as below:
The limitations of our study are due to the fact that spasticity is not determined in the same way for each individual patient. Although spasticity is the same disease, its appearance differs from patient to patient, which is why the botulinum toxin doses cannot be the same for all patients. Another limitation of the study is determined by the fact that spasticity is a syndrome, not a disease, which is constantly dynamic and can be negatively influenced by certain factors: low temperature, emotions, stress, which will lead to an increase in spasticity. Similarly, other limitations are determined by electro stimulation programs and radial shockwave therapy parameters vary depending on the patient’s response to stimulation and spasticity presentation. This is why our study used two spasticity quantification scales. Another limitation could be the number of patients, larger cohorts and more centers would be needed to obtain more data. It is also the reason why there are many discussions in the therapeutic approach in the medical literature. The spasticity of the stroke patient remains an element to be further studied both as a mechanism and as a method of treatment, using complex schemes, therapeutic associations that require time and collaboration from the patient.
Due to these limitations determined by the conditions, the approach of our study is complex and presents the evaluation of the response through several scales, two for spasticity and three for functionality.

Round 2
Reviewer 1 Report
Comments and Suggestions for Authors
Thank you for revising the paper according to my review.
Congratulations on completing your article.
Reviewer 3 Report
Comments and Suggestions for Authors
The clarity of the manuscript has improved after revision. I have no further Comments